# Empirical Mode Decomposition and Hilbert Spectrum for Abnormality Detection in Normal and Abnormal Walking Transitions

**DOI:** 10.3390/ijerph20053879

**Published:** 2023-02-22

**Authors:** Bayu Erfianto, Achmad Rizal, Sugondo Hadiyoso

**Affiliations:** 1School of Computing, Telkom University, Bandung 40257, Indonesia; 2School of Electrical Engineering, Telkom University, Bandung 40257, Indonesia; 3School of Applied Science, Telkom University, Bandung 40257, Indonesia

**Keywords:** human activity recognition, joint, PoseNET, Hilbert Huang Transform

## Abstract

Sensor-based human activity recognition (HAR) is a method for observing a person’s activity in an environment. With this method, it is possible to monitor remotely. HAR can analyze a person’s gait, whether normal or abnormal. Some of its applications may use several sensors mounted on the body, but this method tends to be complex and inconvenient. One alternative to wearable sensors is using video. One of the most commonly used HAR platforms is PoseNET. PoseNET is a sophisticated platform that can detect the skeleton and joints of the body, which are then known as joints. However, a method is still needed to process the raw data from PoseNET to detect subject activity. Therefore, this research proposes a way to detect abnormalities in gait using empirical mode decomposition and the Hilbert spectrum and transforming keys-joints, and skeletons from vision-based pose detection into the angular displacement of walking gait patterns (signals). Joint change information is extracted using the Hilbert Huang Transform to study how the subject behaves in the turning position. Furthermore, it is determined whether the transition goes from normal to abnormal subjects by calculating the energy in the time-frequency domain signal. The test results show that during the transition period, the energy of the gait signal tends to be higher than during the walking period.

## 1. Introduction

Human Activity Recognition (HAR) aims to recognize a series of activities carried out by someone in an environment or area, either closed or open. HAR seeks to understand people’s daily activities by examining insights collected from individuals and their surroundings [1]. Currently, techniques are collected from various IoT sensors embedded in smartphones, wearables, and settings [2]. Human activity recognition (HAR) techniques play an essential role in monitoring daily activities, especially the activities of the elderly, activities investigation, health care, and sports [3]. Another study used smartphones with various embedded sensors for activity recognition [4]. Motion sensors such as accelerometers and gyroscopes are widely used as inertial sensors to identify multiple human physical conditions. In recent research, much work has been conducted regarding introducing human activity [1].

Human activity is the sum of human actions, where action can be defined as the essential and most minor elements of something achieved by humans. Human activity recognition is not a simple process because someone can perform an action periodically. Two activities may have almost similar properties in terms of the similarity of the signal captured from the camera. Therefore, how an activity is carried out may differ from person to person [1]. In their paper, Dang et al. comprehensively discuss sensor-based and sensor-based HAR computer vision [1]. Computer vision-based HAR relies on visual sensing technology, such as cameras or closed-circuit television (CCTV), to record human activity [5]. It does not require sensors, however, depending on the quality of the image obtained. Unlike computer vision, sensor-based HAR will mount a number of sensors on parts of the body so that data that represents human activity can be obtained. Data collected is a series of time series data and parameters. The effect will be analyzed using a statistical and probabilistic approach. Most of the introduction of human activities still do not consider the transition of activities because the duration is shorter than the activity or basic movement in general [6]. Activity transition is a limited event determined by the start time of the lock from one activity to another. On the other hand, in many practical scenarios, such as fitness monitoring systems, determining activity transitions based on kinematic patterns is very important because it is carried out quickly [7]. In terms of human activity recognition and transient activity perception systems, the classification will change slightly, and the absence of a defined activity may result in the wrong type. The previous activity transition approach has been carried out by Ortiz et. al. [6], but using the accelerometer sensor instead of computer vision. The weakness of HAR using an accelerometer sensor is that the sensor is attached to the patient’s body [1]. This form of the sensor is usually embedded in a smartphone [7]. One solution to the barriers to using wearable sensors is video-based HAR [8]. One of the HAR platforms is PoseNET, which can detect the skeleton of the body and body joints known as joints [9]. Creating PoseNET provides a platform for detecting frames and joints, but it is still necessary to process the data to see activity subjects such as gait and so on [10].

This paper proposes empirical mode decomposition with Hilbert spectrum techniques for detecting anomalies in gait pattern. The methods involve transforming vision-based pose estimation of key joints into angular displacement to obtain walking gait patterns (signals). Using the Hilbert Huang Transform to extract walking gait signals can identify information about the activity’s transition, in this case the turning position of the subsequent step. We observe that the energy from the time-frequency domain of gait signals tends to be greater during the transition between two walking activities than during the single walking activity. Therefore, turning information can be used to estimate whether a person has a normal gait pattern or one with an abnormal gait pattern. The proposed method may be used to monitor elderly individuals or people with gait irregularities or diseases such as Alzheimer’s or Parkinson’s.

The remainder of this paper is presented in four sections. Section 2 contains explanations of previous studies closely related to the proposed study. The materials and methods used in this study are presented in Section 3. Section 4 discusses the study’s results, including the energy features for each case and the classification results. A discussion of the results is also presented in this section. The last section contains conclusions, limitations, implications, and challenges for future studies.

## 2. Related Works

Sensor-based human activity recognition is the earliest method used by people. The advantages of HAR-based sensors, especially wearable sensors, are [11]:Wearable sensors can overcome static environmental conditions due to static camera placement.With many sensors used, the accuracy obtained will be higher.Each sensor is intended to obtain specific signals from certain people.Wearable sensors better protect user privacy compared with cameras that can record the user’s face and body.There needs to be a supervisor to observe the video recording.Video signal processing requires more computational challenges.

The weakness of sensor-based HAR is the difficulty in sensor placement, which can cause discomfort to the patient; the number of sensors is not certain depending on the needs, power supply, data transmission, and signal processing [2]. One of the most widely used sensors is the accelerometer and gyroscope [12]. These two sensors are often used for falling detection [13] or detecting respiration activity [14]. Apart from movement, wearable sensors that are often used for HAR, such as body temperature sensors, the Global Positioning System (GPS), heart rate monitors, and others, are intended to monitor the user’s health condition. The data acquired by the sensor is then analysed using signal processing methods and classified.

The study by Nurwulan and Jiang uses accelerometer data for recognition and then analyzes the effect of window selection on HAR [15]. Meanwhile, other researchers explored the impact of smartphone position transitions on acquiring data from sensors [16]. Deep learning is becoming the classification method of choice in today’s HAR. Bhattacharya et al. developed the Ensemble Deep Learning Model for HAR classification based on smartphone sensors [12]. However, some researchers still use traditional machine learning, such as KNN, which has been improved [17].

In addition to the weaknesses previously described, video-based HAR has several advantages. It does not interfere with movement, only one type of signal is taken, and it does not require multiple installations [18]. Generally, three types of vision sensors are often used: deep sensors, visible spectrum sensors, and infrared sensors [19]. The deep sensor commonly used is the Microsoft Kinect, which can extract the skeleton from images [20]. RGB cameras usually require more intensive image processing methods such as PoseNet [9,21]. Meanwhile, for infrared, you can use infrared break beam sensors, ultrasonic sensors, and passive infrared sensors [19]. PoseNet plays an important role in developing vision-based HAR [9]. PoseNet uses deep learning to generate 3D spatial coordinates of joint human skeleton poses. Examples of vision-based HAR applications include the detection of falls in the elderly [22] or the detection of the subject’s pose [23].

## 3. Materials and Methods

The proposed system block diagram is shown in Figure 1. In this study, the Openpose pose framework was used to extract the knee joint from the subject. Carnegie Mellon University (CMU) developed OpenPose, a supervised convolutional neural network based on Caffe, for real-time multi-person 2D pose estimation [24]. It can estimate the posture of human body movements, facial expressions, and finger movements. It has an excellent recognition effect and a fast recognition speed, making it suitable for single- and multiple-user settings. After obtaining the knee joint, an analysis was carried out using empirical mode decomposition (EMD) and the Hilbert Huang Transform (HHT) to detect gait patterns and turning events. Details of each stage are described in the sections below.

Figure 1 shows the three main stages in this study, including preprocessing, signal processing, and training and testing. At the preprocessing stage, the video dataset is taken for pose estimation, in this case using PoseNet. The output from the pose estimation engine is the position of the key joints and the artificial skeleton in each video frame, which is overlaid with the original video. However, for the next process, only the coordinates of the key joints and the artificial skeleton are used. In the second stage, signal processing is carried out by extracting joints and converting them into joint coordinates and joint kinematic angles (flexion and extension). The result of this extraction is in the form of gait activity signals. Furthermore, gait activity signals are extracted using the Hilbert Huang Transform, with the initial stage being EMD processing to obtain the intrinsic mode function (IMF) signal components [8]. In the case of this study, the 6th IMF signal is used as a feature for classification. The entire IMF is also processed by the Hilbert Transform algorithm to obtain the energy spectrum from the gait activity signal. The energy spectrum value, along with the subsequent IMF, is used as a feature for the K-Nearest Neighbor (KNN) classifier. The results of the classification of the training dataset and testing dataset are grouped into Normal Walking activity, Parkinson’s disease activity, and other disordered walking activity classes.

### 3.1. Dataset

The data used in this study are video recordings where the subject performs walking and turning activities. Videos are taken from several YouTube channels, one of which is https://www.youtube.com/c/MissionGait, accessed on 1 August 2022, with different subjects and varied backgrounds. The video used in this study is in mp4 format with a frame rate of 30 fps. Figure 2 shows an example of the video analyzed in this study. A total of 14 videos were processed in this study, including normal and abnormal walking.

In Figure 2 and Figure 3, you can see an example of a video taken from the Mission Gate dataset. In the picture, the left is the original video, and the right is the overlay video between the original video and the artificial skeleton and joints resulting from pose estimation at the preprocessing stage (see the method in Figure 1). From a total of 15 datasets used, the pose estimation engine succeeded in overlaying the original video with artificial skeletons and joints. Only the joints related to the lower limb are extracted for gait activities, namely the knee and hip joints, and the ankle joint is extracted so that an artificial skeleton can be constructed that connects the joints so that the angles between the joints can be calculated.

The video dataset used for this experiment can be seen in Table 1. Mission Gait provides a series of videos demonstrating real-life individuals with gait disorders. Thus, this video dataset can serve as a dataset for research, therapist practice, and telemedicine applications for early diagnosis. In this study, 15 datasets represented normal walking activities and some gait disorder activities, as seen in Table 1. The video dataset used for this experiment came from Mission Gait, with screenshots as shown in Table 1. Mission Gait provides a series of videos demonstrating real individuals with gait disorders. Thus, this video dataset can serve as a research dataset, therapist practice, and telemedicine for initial diagnosis. In this study, 13 datasets represented normal walking activities and some gait disorder activities, as seen in Table 1.

Some disorder terminology is taken from WebMD [12]. The several decoded categories of the video dataset, according to those on the mission gait site are as in Table 2.

In general, how the gait moves will provide early clues as to the underlying cause of the gait problem. This can help the doctor or therapist diagnose the problem and plan therapy. Each type of gait disorder has variations, and no two people will have the same symptoms. Thus, the dataset will produce gait patterns with the general characteristics extracted in this study using signal processing methods.

### 3.2. Empirical Mode Decomposition (EMD)

Huang et al. introduced Empirical Mode Decomposition (EMD) in 1998 as a new and effective tool for analyzing non-linear and non-stationary signals [25,26]. A complicated and multiscale signal can be adaptively decomposed into the sum of a finite number of zero-mean oscillating components known as Intrinsic Mode Functions using this method (IMF). 

EMD’s specific steps are as follows:

Step 1: Determine the maximum and minimum signal × values (*t*).

Step 2: To obtain the highest possible value *e*_max_(t), use the cubic spline interpolation function to fit the upper envelope; use the cubic spline interpolation function to fit the lower envelope *e*_min_(t) for the minimum value.

Step 3: Calculate *m*(*t*):(1)$$m\left(t\right)=\frac{e_{\max}\left(t\right)-e_{\min}\left(t\right)}{2}$$

Step 4: Calculate modal function *c*(*t*):(2)c(t)=x(t)−m(t)

Step 5: If *c*(*t*) satisfies IMF condition, then *c*(*t*) is an IMF component, and the original signal becomes *x_n_*_+1_(*t*), then:(3)$$x_{n+1}\left(t\right)=x\left(t\right)-c\left(t\right)$$

If *c*(*t*) does not satisfy IMF condition, go back to step 3.

Step 6: The residual component *r*(*t*) is reserved and the decomposition is complete when the result signal has less than two extremum locations. The original signal is divided into n IMFs, with *r*(*t*) being the residual component.
(4)$$x\left(t\right)=\sum_{i=1}^{N}c_{i}\left(t\right)+r\left(t\right)$$

### 3.3. Hilbert Spectrum

Hilbert spectral analysis is a signal analysis technique that uses the Hilbert transform to calculate the instantaneous frequency of signals [27].
(5)$$\omega =\frac{d\theta }{dt}$$

After applying the Hilbert transform to each signal, we can express the data as in Equation (6).
(6)$$X\left(t\right)=\sum_{j=1}^{n}a_{j}\left(t\right)exp\left(i\int \omega_{j}\left(t\right)dt\right)$$

As a function of time, this equation gives the amplitude and frequency of each component. It also allows us to represent the amplitude and instantaneous frequency as time functions in a three-dimensional plot, with the amplitude contoured on the frequency-time plane. The Hilbert amplitude spectrum, or simply Hilbert spectrum, is the frequency-time distribution of the amplitude. The Hilbert spectral analysis method is an essential component of the Hilbert-Huang transform [28].

## 4. Results and Discussion

### 4.1. Detection Confidence

Body pose estimation, in this case using machine learning-based pose estimation, works based on a specific confidence value, whether a landmark (joint) is detected or no activity is taking place. This argument will set the threshold value of the confidence level and range from [0.0, 1.0], i.e., the minimum confidence level is 0, and the maximum confidence level is one. By default, the value is 0.5. Thus, the pose estimation engine will determine whether the joint is detected based on the given confidence level. In Figure 4, it can be seen the evolution of the confidence value (in percent) during the pose estimation process.

The video used in this example is from the Case Study 10 Dataset, with the direction of activity running from right to left of the video. In the 4th second, the confidence value goes to the lowest point due to occlusion, or the object is covered so that the pose estimator detects joints with a low level of visibility (low detection confidence). Overall, the detection confidence of all dataset files can be mapped using the probability density function for both the left and right knee, as seen in Figure 4. The lowest mean value of detection confidence is around 90%, and the highest mean is about 98%. The closer the detection confidence value is to 1, the more accurately the detected joints are detected, so when it is used to define poses per video frame, the specified poses are also more accurate. Of course, this will affect the tracking results of an angle formed from various artificial joints and seconds.

Figure 5 displays the probability density function of detection confidence for all datasets used. It can be seen that the right knee produces a higher probability density than the left knee. This will be a consideration in selecting features for the classification of abnormalities.

### 4.2. Gait Signal Based on Video Dataset 

Figure 6 shows the gait signal in the form of angular values taken from the right and left knees of a person walking normally. The graph shows the regular pattern of angular values throughout the observation time. Meanwhile, Table 3 shows the gait signal in several cases, represented by Case Study 10 for normal and Case Studies 6 and 5 for subjects with Parkinson’s and diabetic neuropathy. In case study 6, subjects with Parkinson’s produced a significant difference in turn times (4th to 14th seconds).

### 4.3. Feature Extraction Using Hilbert Huang Transform

HHT is a combination of two methodologies [29], namely, empirical mode decomposition (EMD) and Hilbert transform (HT). In the first step, the input signal will be decomposed into different components using EMD, which are called intrinsic mode functions (IMFs) [30]. In the second step, the Hilbert spectrum is obtained by conducting HT over IMF. HHT can adaptively decompose IMF and also has better time-frequency resolution when compared with the wavelet transform (WT) and short-time Fourier transform (STFT) [31]. Figure 7 shows IMF-1 through IMF-6 resulting from the gait signal decomposition case study 10. Meanwhile, Figure 8 shows the original gait signal and IMF-5, which represent two different activities. When the subject performs the turning activity, the signal value from IMF-5 drops significantly.

Figure 9 shows the Hilbert energy spectrum of the right and left knee from case studies 10 and 6. The energy is plotted over the time span, representing the two different activities. When the subject performs a turning activity, it produces a smaller energy value than walking activity. However, in subjects with Parkinson’s, low energy occurs with a longer duration, indicating that the subject has difficulty turning as depicted in Figure 10.

Based on the features formed, we created a class definition as display in Table 4.

### 4.4. Classification Result

Table 5 shows the accuracy of the three gait class classifications using the random forest classifier (RF). The parameters used to measure its performance are the True Positive Rate (TPR) and False Negative Rate (FNR). Mathematically, the TPR and FNR are expressed as in Equations (7) and (8) [32].
(7)$$TPR=\frac{TP}{FN+TP}$$
(8)$$FNR=\frac{FN}{FN+TP}=1-TPR$$

From Table 4, it can be seen that for NW, most of the TPR values > FPR, meaning that more NW are recognized correctly as NW compared with those incorrectly recognized. Meanwhile, the OW tends to have a TPR value < FPR, which means that OW is more likely to be misrecognized. Meanwhile, due to the small amount of data for PD, it can be seen that for data 6, the TPR reaches 100% while the other data is TPR < FNR. In general, the results of the proposed method have yet to yield sufficiently good results; this is due to the results only showing when a transition occurs while running. These results have yet to be further analyzed to determine the difference between the transition signals under certain disease conditions and under normal conditions. From this initial observation, it was found that one of the differences that can be used is the duration of the transition process. In this study, the dataset’s limitations are also one reason the accuracy tends to be low. In addition, this study used only the left and right knee joints as signal sources. Adding the number of joints analyzed is also a potential area for further research. Despite the low accuracy, the proposed method has the advantage of being easy in the data acquisition process because it does not require a sensor attached to the subject’s body [21]. This method has the potential to be applied to in-home rehabilitation for the elderly.

## 5. Conclusions

This study proposes a method for detecting abnormalities in walking transitions using the EMD and Hilbert spectra. The input data is in the form of a walking video of the subject extracted using PostNet. In this study, only the left knee was analyzed because it was considered to represent the subject’s movement. A person with an abnormal gait has a longer turning duration than normal. In the case of people with Parkinson’s, it produces a turn time about 10 s, while in normal people it takes 2 s. From the experiment, it was found that the location of the walking transition can be seen using the Hilbert spectrum. Hilbert’s energy spectrum shows that when the subject performs the turning activity, it produces a smaller energy value than the walking activity. However, in subjects with Parkinson’s, low energy occurs with a longer duration, indicating that the subject has difficulty turning. Furthermore, the classification of cases of normal walking (NW), other disorder walking (OW), and Parkinson’s Disease (PD) was carried out. The classification results show that NW produces a higher TPR > FPR, which means that more NW are classified correctly. Whereas OW tends to have a TPR < FPR value, which means OW is more likely to be recognized incorrectly. Meanwhile, due to the small amount of PD data, it can be seen that for case study 6, the TPR reached 100% while the other data was TPR < FNR. A future study needs further analysis to determine the difference between transition signals under certain disease conditions and under normal conditions. In this study, the limited dataset is also one of the causes of the low accuracy. Exploration to extract the time of walking transitions is also the focus of the following research. The method proposed in this study has the potential to be applied to home rehabilitation for the elderly.

## Figures and Tables

**Figure 1 ijerph-20-03879-f001:**
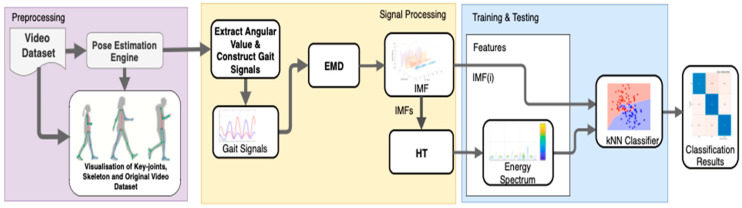
Diagram block of proposed method.

**Figure 2 ijerph-20-03879-f002:**
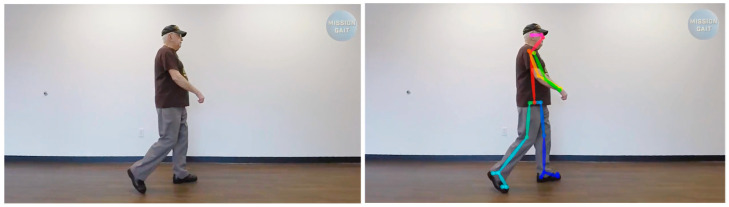
Walking activity Case Study 5. (**Left**): Original Video. (**Right**): Overlaying Pose Estimation results into original video.

**Figure 3 ijerph-20-03879-f003:**
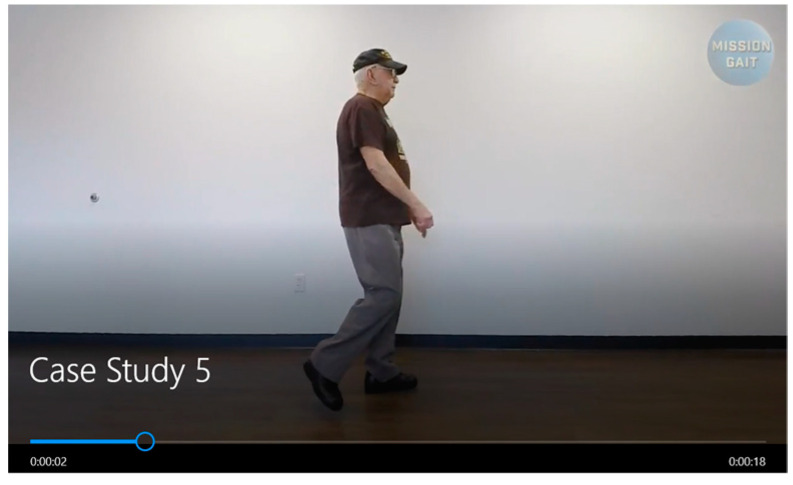
Video showing someone walking.

**Figure 4 ijerph-20-03879-f004:**
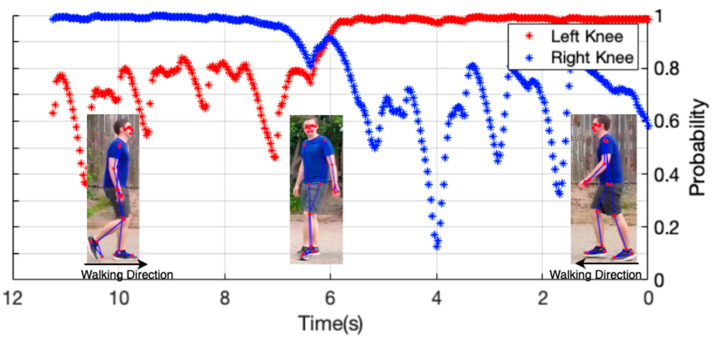
Detection confidence during walking and turning activities.

**Figure 5 ijerph-20-03879-f005:**
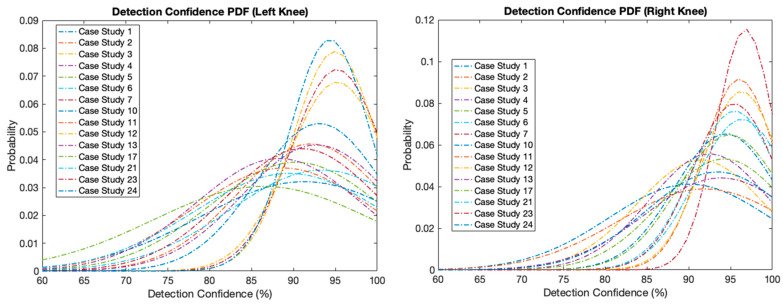
Probability density function of detection confidence for all datasets used.

**Figure 6 ijerph-20-03879-f006:**
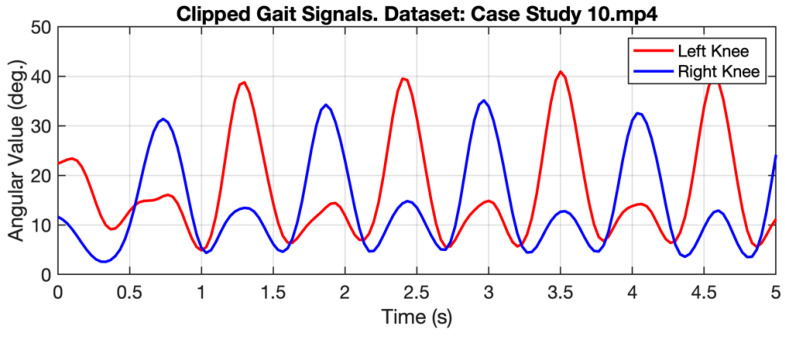
Example of Gait Signal obtained from pose estimation.

**Figure 7 ijerph-20-03879-f007:**
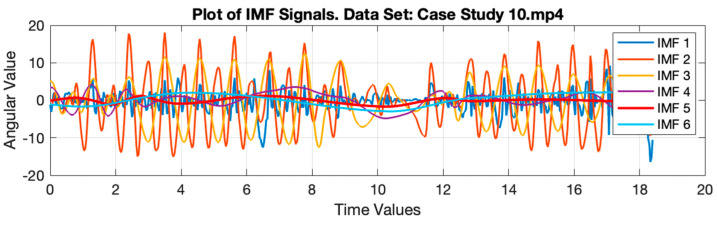
Example of IMF Signals decomposed from Gait Signals using IMF 6.

**Figure 8 ijerph-20-03879-f008:**
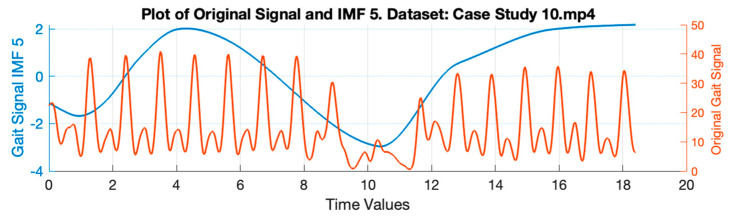
Example of IMF 5 Signal with Original Gait Signals.

**Figure 9 ijerph-20-03879-f009:**
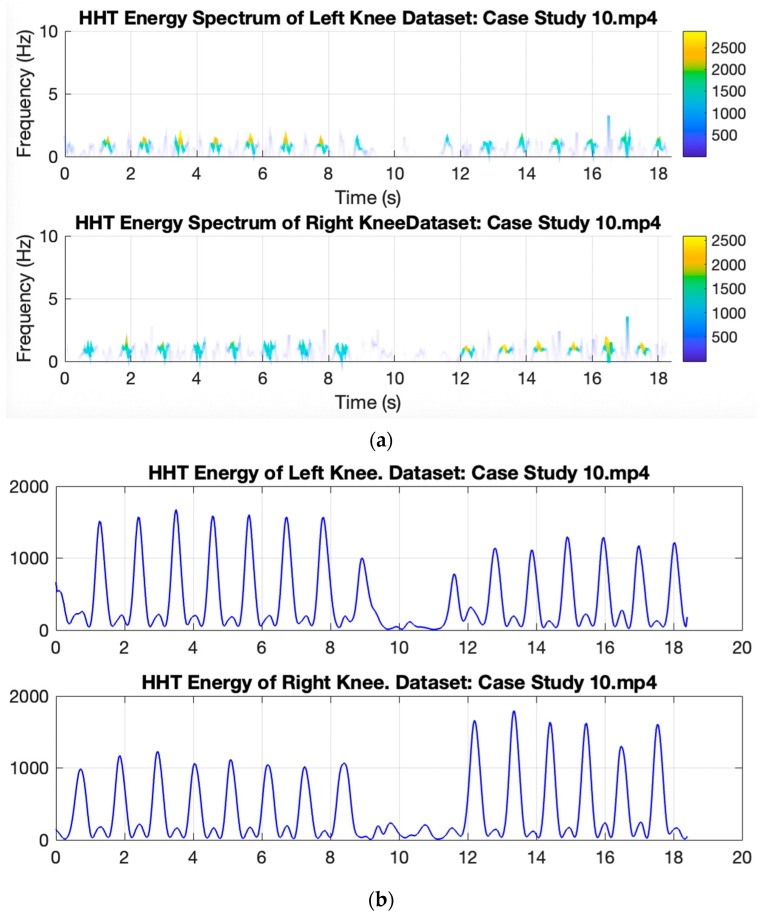
Example of Hilbert spectrum from normal gait signal from case study 10. (**a**) HT Energy Spectrum. (**b**) Energy extracted from HHT Energy Spectrum.

**Figure 10 ijerph-20-03879-f010:**
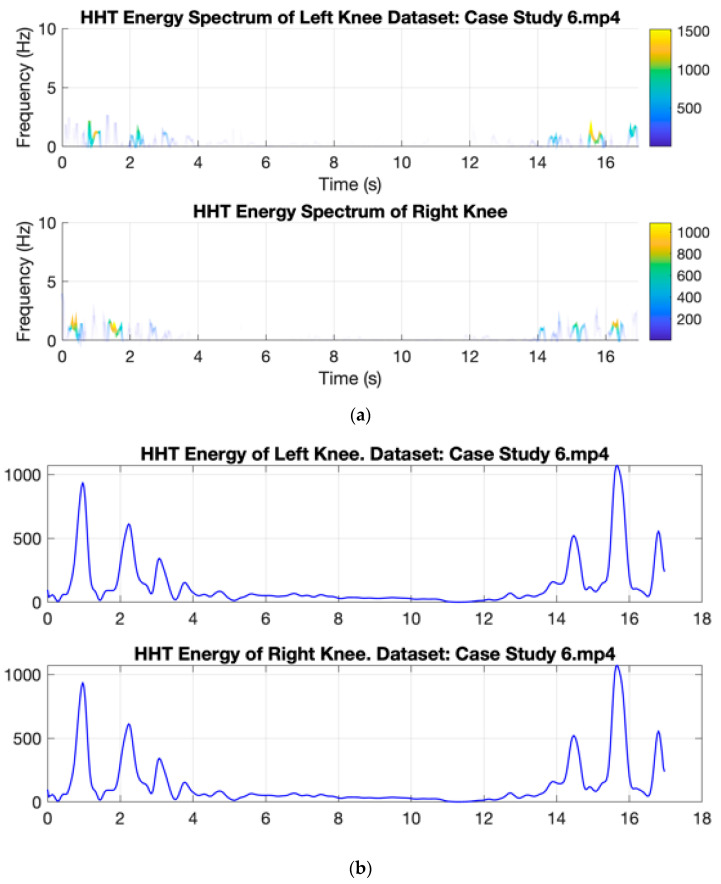
Example of Hilbert spectrum from Parkinson Disease Walking Activity from case study 6. (**a**) HT Energy Spectrum. (**b**) Energy extracted from HHT Energy Spectrum.

**Table 1 ijerph-20-03879-t001:** Video Dataset.

Case Study						
1	2	3	4	5	6	7
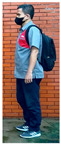	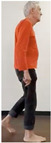	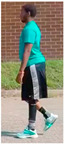	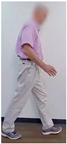	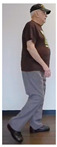	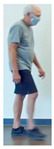	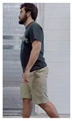
N	LLD	KA	PD2	DN	PD 1	N
Case Study						
10	11	12	13	17	21	23
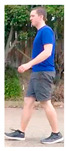	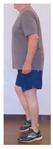	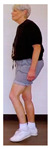	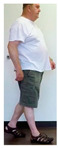	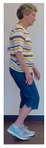	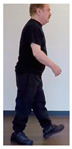	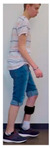
N	N	MS	HP	CH	PD3	BI

**Table 2 ijerph-20-03879-t002:** Video Dataset Category.

Dataset	Video Duration (Second)	Initial Direction	Status Gait
Case Study 1	13	Right to Left	N: Normal, normal walking gait taken from author
Case Study 2	16	Left to Right	LLD: Leg Length Discrepancy
Case Study 3	25	Right to Left	KA: Knee Amputee is amputation from knee below
Case Study 4	13	Left to Right	PD2: Parkinson Disease (Mild)
Case Study 5	20	Left to Right	DN: Diabetic Neuropathy is a motor nerve disorder caused by diabetes.
Case Study 6	17	Left to Right	PD1: Parkinson Disease (Severe)
Case Study 7	18	Right to Left	Normal (Video from mission gait)
Case Study 10	20	Right to Left	Normal (Video from mission gait)
Case Study 11	18	Right to Left	Normal (Video from mission gait)
Case Study 12	20	Right to Left	MS: Multiple Sclerosis is a movement disorder due to communication disorders between the brain and the motor system of the body
Case Study 13	21	Left to Right	HP: Hemiplegic, motoric disorder as common long-term consequence of stroke
Case Study 17	29	Left to Right	CH: Chronic Hemiparetic
Case Study 21	22	Left to Right	PD3: Parkinson Disease (Moderate)
Case Study 23	11	Left to Right	BI: Brain Injury
Case Study 24	14	Left to Right	DS: Dystonia is movement disorder that causes the muscles to contract involuntarily

**Table 3 ijerph-20-03879-t003:** Gait value in normal and abnormal cases.

	Gait Signals
Case Study 10	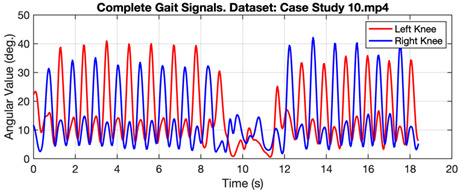
Case Study 6	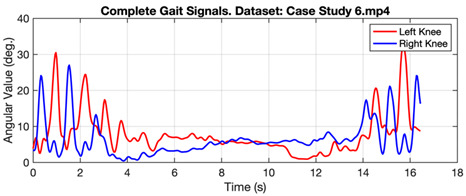
Case Study 5	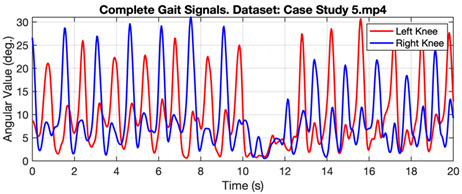

**Table 4 ijerph-20-03879-t004:** Class definition of each gait signals. NW: Normal Walking, OW: Other disorder walking, PD: Parkinson Disease.

Dataset	Class
Case Study 1	NW
Case Study 2	OW
Case Study 3	OW
Case Study 4	PD
Case Study 5	OW
Case Study 6	PD
Case Study 7	NW
Case Study 10	NW
Case Study 11	NW
Case Study 12	OW
Case Study 13	OW
Case Study 17	OW
Case Study 21	PD
Case Study 23	OW
Case Study 24	OW

**Table 5 ijerph-20-03879-t005:** Classification Results. NW: Normal Walking. OW: Other Disorder Walking. PD: Parkinson Disease Walking. TPR: True Positive Rate. FNR: False Negative Rate.

	NW		OW		PD	
	TPR	FNR	TPR	FNR	TPR	FNR
Case Study 1 [NW]	0.618	0.382	0	0	0	0
Case Study 2 [OW]	0	0	1	0	0	0
Case Study 3 [OW]	0	0	0.311	0.688	0	0
Case Study 4 [PD]	0	0	0	0	0.14	0.86
Case Study 5 [OW]	0	0	0.46	0.54	0	0
Case Study 6 [PD]	0	0	0	0	1	0
Case Study 7 [NW]	0.759	0.241	0	0	0	0
Case Study 10 [NW]	1	0	0	0	0	0
Case Study 11 [NW]	0.256	0.744	0	0	0	0
Case Study 12 [OW]	0	0	0.328	0.672	0	0
Case Study 13 [OW]	0	0	0.481	0.519	0	0
Case Study 17 [OW]	0	0	0.344	0.656	0	0
Case Study 21 [PD]	0	0	0	0	0.266	0.733
Case Study 23 [OW]	0	0	0.452	0.548	0	0
Case Study 24 [OW]	0	0	0.379	0.621	0	0

## Data Availability

Not applicable.

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
