# Peer review of "Empirical Mode Decomposition and Hilbert Spectrum for Abnormality Detection in Normal and Abnormal Walking Transitions"

_ijerph, 2023, doi:10.3390/ijerph20053879_

Round 1

Reviewer 1 Report

I believe that the scientific level and originality of the article should be increased.

The title of the paper does not reflect the content very well. Even the authors say that the gait abnormalities are not visible, and the duration cannot be determined.

Moreover, the conclusions do not fully reflect the content of the research

Author Response

Response to Reviewer 1 Comments

Point 1: I believe that the scientific level and originality of the article should be increased.

 Response 1: Thank you for the valuable comment, we have highlighted the novelty, originality, and contribution of this study in the introduction section.

 Point 2: The title of the paper does not reflect the content very well. Even the authors say that the gait abnormalities are not visible, and the duration cannot be determined.

 Response 2: Thank you for the valuable comment, we have added a more detailed explanation of the purpose of this study and the method used in the Introduction section (last paragraph). A comparison of the duration for normal and abnormal gait has also been presented in section 3.2 and the conclusions.

Point 3: Moreover, the conclusions do not fully reflect the content of the research

Response 3: Thank you for concerning this issue, we have made a more comprehensive conclusion regarding the research objectives, the proposed methods, and the results that have been achieved. Limitations, implications, and future studies have also been presented.

Reviewer 2 Report

This paper provides a new method for gait normally and abnormally detection, which uses empirical mode decomposition and Hilbert spectrum to process data. On the one hand, this method uses visual data sets to avoid using multiple sensors to collect data. On the other hand, the method can effectively detect gait abnormalities by transforming keys-joints and skeletons into the angular displacement of walking gait patterns. There are some problems, which should be solved before it is considered for publication.

1. Relevant research background needs to be supplemented in INTRODUCTION. There is a lack of methods for anomaly detection of existing video data to reflect the advantages of this method.

2. The formula (1) in line 167 and the formula (2) in line 169 are repeated. Please confirm whether the expression in c (t) is written incorrectly.

3. It is suggested that "emax (t)" in line 163 be replaced by "emax (t)", which is expressed in subscript form. This suggestion is used for the whole paper.

4. The left figure of Figure 6 is a bit vague. Please consider replacing it with a clearer one.

5. The "FNS" in line 266 and the expression "FNR" in line 270 sound incorrect.

6. The formula in line 277 should be FNR instead of FPR, which does not match the data in Table 4. It is recommended to carefully sort out the relationship between TPR, TNR, FNR and FPR.

7. The significance of this paper is not expound sufficiently. The author need to highlight this paper's innovative contributions.

Author Response

Response to Reviewer 2 Comments

Point 1: Relevant research background needs to be supplemented in INTRODUCTION. There is a lack of methods for anomaly detection of existing video data to reflect the advantages of this method.

Response 1: 

We have added sentences to explain the lack of the previous method in the Introduction. See lines 61-64. Thank you for your valuable suggestion.

Point 2: The formula (1) in line 167 and the formula (2) in line 169 are repeated. Please confirm whether the expression in c (t) is written incorrectly.

Response 2:

Thank you for the correction; we wrote the wrong formula (2); it should be like the formula below. We have corrected it in the manuscript.  

Point 3: It is suggested that "emax (t)" in line 163 be replaced by "emax (t)", which is expressed in subscript form. This suggestion is used for the whole paper.

Response 3.

We apologize for our negligence in writing the equation. All emax(t) and emin(t) have been fixed according to the reviewers' suggestions.

Point 4: The left figure of Figure 6 is a bit vague. Please consider replacing it with a clearer one.

Response 4.

We have split Fig. 6 into Fig. 6 and Fig. 7 and resized them to a bigger and clearer view.

Point 5: The "FNS" in line 266 and the expression "FNR" in line 270 sound incorrect.

Response 5.

Sorry for our negligence. The correct one is the False Negative Rate (FNR). All writing errors (FNS and FPR on Eq (8)) have been corrected according to the corrections from the reviewers.

Pont 6: The formula in line 277 should be FNR instead of FPR, which does not match the data in Table 4. It is recommended to carefully sort out the relationship between TPR, TNR, FNR, and FPR.

Response 6.

Thanks for the correction; Eq. 8 has been fixed too. Table 4 has been corrected and checked for errors.

Point 7: The significance of this paper is not expound sufficiently. The author need to highlight this paper's innovative contributions.

Response 7:

Thank you for the valuable comment, we have added the paper contribution in the introduction (last paragraph)

Reviewer 3 Report

In the introduction, there are some stand alond sentences, please revise them:

1. Page 1 line 36, "A smartphone that incorporates various sensors"

2. Page 2 line 49 "Unlike the sensor-based HAR..."

Revise for clarity:

2. Page 2 line 42: "Introduction to human activity..."

Please add the name of the authors in the following in text reference:

Page 2 line 60: "The previous activity transition....carried out by (6)

Author Response

Response to Reviewer 3 Comments

Point 1: In the introduction, there are some stand alone sentences, please revise them:

  1. Page 1 line 36, "A smartphone that incorporates various sensors"

  1. Page 2 line 49 "Unlike the sensor-based HAR..."

 Response 1: Thank you for the correction, we have fixed the issue in page 1 line 36 and page 2 line 49.

Point 2: Please clarify Page 2 line 42: "Introduction to human activity..."

 Response 2: Thank you for the comment, we have changed the sentence to make it clearer.

Point 3: Please add the name of the authors in the following in text reference: Page 2 line 60: "The previous activity transition....carried out by (6)

Response 3: Thank you for concerning this issue, we have added the author’s name in the sentence.

Reviewer 4 Report

This study proposes a way to detect abnormalities in gait using empirical mode decomposition and the Hilbert spectrum by transforming keys-joints and skeletons from vision-based pose detection into the angular displacement of walking gait patterns (signals). 

The work is interesting and original. 

The followings can be done to increase the quality of the paper: 

1- The "Related Work" section is missing. A new section may be added between "Introduction" and "Section 2".  

Some parts of the "Introduction" section can be moved to the new section. 

In the reference list, there is no any paper published in 2022 and 2023.  

I suggest the authors citing the most recent papers (especially published in 2022 and 2023).

2- The TPR values given in Table 4 are low (64.2 52.2 20.7 42 45.5 etc.). 

KNN may not provide a good classification accuracy. Maybe a different classification algorithm (i.e., Random Forest) can be used to increase the results.    

3- The organization of the paper (the structure of the manuscript) may be written at the end of the "Introduction" section. 

For example: "Section 2 presents ... Section 3 gives ...." 

4- Some abbreviations are used in the text without giving their expansion.   

For example; CCTV 

The authors can write that "these abbreviations stand for what".

Author Response

Response to Reviewer 4 Comments

Point 1: 1- The "Related Work" section is missing. A new section may be added between "Introduction" and "Section 2".  Some parts of the "Introduction" section can be moved to the new section. In the reference list, there is no any paper published in 2022 and 2023.  I suggest the authors citing the most recent papers (especially published in 2022 and 2023).

Response 1: Thank you for the valuable comments. We have added related works with the latest references in the new section. We have also added references published in 2022.

Point 2: 2- The TPR values given in Table 4 are low (64.2 52.2 20.7 42 45.5 etc.). KNN may not provide a good classification accuracy. Maybe a different classification algorithm (i.e., Random Forest) can be used to increase the results.   

Response 2: Thank you for the suggestions. We have simulated another classifier algorithm, namely the Random Forest with the results as presented in the section 4.4 Table 4. RF has better classification result compare to KNN.

Point 3: The organization of the paper (the structure of the manuscript) may be written at the end of the "Introduction" section. For example: "Section 2 presents ... Section 3 gives ...."

Response 3: Thank you for the suggestion. We have added the structure of the manuscript at the end of the introduction.

Poinr 4: Some abbreviations are used in the text without giving their expansion. For example; CCTV

Response 3: Thank you for concerning this issue. We have re-checked some of the abbreviations and given them their expansion.

Round 2

Reviewer 2 Report

My concerns have been addressed.